# Circadian Interventions in Preclinical Models of Huntington’s Disease: A Narrative Review

**DOI:** 10.3390/biomedicines12081777

**Published:** 2024-08-06

**Authors:** Derek Dell’Angelica, Karan Singh, Christopher S. Colwell, Cristina A. Ghiani

**Affiliations:** 1Department of Psychiatry and Biobehavioural Sciences, Semel Institute for Neuroscience and Human Behaviour, David Geffen School of Medicine, University of California Los Angeles, Los Angeles, CA 90024, USA; ddellangelica@mednet.ucla.edu (D.D.); ksingh1370@gmail.com (K.S.); ccolwell@mednet.ucla.edu (C.S.C.); 2Department of Pathology and Laboratory Medicine, David Geffen School of Medicine, University of California Los Angeles, Los Angeles, CA 90024, USA

**Keywords:** Huntington’s Disease, neurodegenerative disorder, preclinical models, circadian rhythms, circadian medicine, circadian interventions, sleep

## Abstract

Huntington’s Disease (HD) is a neurodegenerative disorder caused by an autosomal-dominant mutation in the huntingtin gene, which manifests with a triad of motor, cognitive and psychiatric declines. Individuals with HD often present with disturbed sleep/wake cycles, but it is still debated whether altered circadian rhythms are intrinsic to its aetiopathology or a consequence. Conversely, it is well established that sleep/wake disturbances, perhaps acting in concert with other pathophysiological mechanisms, worsen the impact of the disease on cognitive and motor functions and are a burden to the patients and their caretakers. Currently, there is no cure to stop the progression of HD, however, preclinical research is providing cementing evidence that restoring the fluctuation of the circadian rhythms can assist in delaying the onset and slowing progression of HD. Here we highlight the application of circadian-based interventions in preclinical models and provide insights into their potential translation in clinical practice. Interventions aimed at improving sleep/wake cycles’ synchronization have shown to improve motor and cognitive deficits in HD models. Therefore, a strong support for their suitability to ameliorate HD symptoms in humans emerges from the literature, albeit with gaps in our knowledge on the underlying mechanisms and possible risks associated with their implementation.

## 1. Introduction

Huntington’s disease (HD) is a neurodegenerative disorder characterized by a triad of motor, cognitive, and psychiatric declines. The cause of HD is genetic, namely an autosomal-dominant expansion of the CAG polyglutamine repeat within the protein-coding region of the HTT (huntingtin) gene providing a toxic gain of function that results in the dysfunction and death of neurons. The mean onset of diagnosis is around 45 years old, with symptoms of neuronal decline often preceding the diagnosis and progressing inexorably and fatally [1]. Pre-symptomatic and symptomatic HD individuals often lament disturbances in their sleep/wake cycles. The mechanisms underlying sleep/wake deficits in HD are poorly understood; however, there is a great deal of overlap in the non-motor symptoms experienced by HD patients and the consequences of sleep/wake cycle disruptions, raising the possibility that, perhaps, circadian malfunctioning could be an integral part of the disease. Hence understanding the possible underlying role of specific circadian or circadian-regulated genes in HD may aid in the development of targeted interventions to ameliorate or just delay disease presentation and progression. Evidence from preclinical models of HD have shown the effectiveness of circadian-based interventions in ameliorating sleep/wake cycles, motor dysfunctions, slowing disease progression and extending lifespan [2,3,4].

It is difficult to determine whether and how the disease alters the circadian timing system in humans and vice versa, however, animal models have provided critical insights. Several models have been created in rodents, flies, as well as in larger mammals; each recapitulates some of the HD dysfunctions, including perturbed sleep/wake cycles [5,6,7,8,9,10]. ‘Malfunctioning’ of the central clock, the suprachiasmatic nucleus (SCN), has been reported in the R6/2 mouse model, likely due to the reduction in the levels of vasoactive intestinal polypeptide (VIP) and its type 2 receptor (VPAC2), a major player in sleep/wake cycle regulation as well as in genes of the molecular clock [11,12]; while a reduction in the Nissl-defined size of the male, but not female, SCN was shown in the BACHD model [13]. Additionally, as suggested by studies in rodent models, such as the Q175 and BACHD mice, the reduced daytime electrical output from the master circadian clock could be a fundamental deficit contributing to disordered rhythms in HD [14,15]. Most of these models also show reduced sleep, altered rhythms in locomotor activity, and autonomic nervous system (ANS) dysfunctions [16]. At least one study examined post-mortem structural changes in the SCN of individuals affected by HD, evidencing a reduction in vasoactive intestinal peptide (VIP)- and vasopressin (AVP)-expressing cells [17].

The anterior hypothalamus houses the master circadian timing system, the SCN, which orchestrates and harmonizes whole-body functional/physiological rhythms [18,19,20]. Rhythms in neural activity in the SCN are driven by a cell autonomous molecular feedback loop [21,22]. The core clock genes interact in a complex feedback loop, initiating waves of transcription that drive circadian rhythms in almost all neural and peripheral organ cells (Figure 1) with a periodicity close to 24 h [23,24,25]. Several genes and proteins involved in physiological processes, such as metabolism, hormone production, and immune functions, are under the control of such a system and display circadian fluctuations. Overall, the circadian system is a complex network of genes, proteins and biological processes that work together to regulate the many physiological functions and keep the body internal clock in sync with the outside world [26,27,28]. Any interference with the organization and synchronization of this timing system has deleterious effects on health, triggering maladies, including cancer and diabetes, or exacerbating existing malconditions (Figure 1) [25,29]. Hence, disruptions in the circadian timing system might not just be a consequence but an integral component of HD pathophysiology as well as other neurodegenerative disorders.

It is of note, that while in humans no clear sex differences have been reported in HD and a consensus is still lacking [30,31,32], rodent models display clear sex differences in sleep/wake disruptions, with the males being more vulnerable and with an earlier onset [13,33] as well as in motor function and other behaviours [34,35,36,37]. Hence, sex divergency may be of great help to understand the underlying mechanisms of sleep/wake alteration in HD and identify possible interventional targets.

Currently, there is no cure for HD [1]. Therefore, for these individuals, improvements in circadian rhythms and sleep quality might be vital to delay disease presentation and slow its progression, e.g., lessening the neurodegenerative events as well as the declines in motor and non-motor dysfunctions. Increasing evidence from preclinical models is strongly suggesting that interventions aimed at restoring sleep/wake cycles may be a way forward (Figure 2). For instance, sleep slow-wave activity is directly associated with the ability of the astroglial-mediated glymphatic system to clear neural waste, including protein aggregates: a hallmark of many neurodegenerative disorders, including HD [38]. Changes and deficits in sleep slow-wave activity have been reported in in the R6/2 mouse model [12,39]. Moreover, while disturbed sleep might be associated with impaired clearance of neural waste and a more rapid decline in motor and cognitive function, higher sleep quality may ameliorate glymphatic drainage, slowing the decline associated with neurodegenerative disorders. It is noteworthy to emphasize that several of the body clearance systems, including the glymphatic system and autophagy are both under circadian control and display endogenous rhythms [40,41]. Several neurodegenerative disorders present with impaired autophagy, which seems to be repristinated by some circadian-based interventions [42,43,44].

In the next sections, we will review the application of different circadian-based interventions in preclinical models of HD and then provide insights into their potential effectiveness and suitability for translation into clinical practice.

## 2. Preclinical Evidence

### 2.1. Environmental Enrichment

Environmental enrichment (EE) can be understood as modifications in the living conditions of the animals with the aim of providing activities that enhance their motor or cognitive function. In HD, the benefits of EE in preclinical models are mostly credited to morphological and neuroplastic changes in the brain; including increased expression of proteins [45], neurotrophic and growth factors, changes in genetic and epigenetic factors, in the hypothalamic-pituitary-adrenal (HPA) axis functions and hippocampal cytoarchitecture [46,47]. In animal models, including the R6/2 mouse model of HD and developing pigs, it has been demonstrated that EE can advance circadian phase, delay circadian disintegration, and improve synchronization of activity rhythms to the light-dark (LD) cycles [48,49,50]. Exercise, in particular, can facilitate and promote entrainment of free-running behavioural rhythms and alignment to the LD cycle while influencing the duration of endogenous periods in humans as well as in both diurnal and nocturnal rodents [51]. Moreover, exercise can alleviate age-related circadian desynchrony, as well as influence spontaneous firing rate rhythms and gene expression in the SCN [52,53]. It is possible that these circadian-related benefits are influenced by morphological and neuroplastic changes in the brain, for instance, Manno and colleagues [50] reported that circadian advancements elicited by EE in naïve C57BL/6j mice were associated with increased CA1 neuronal spiking and functional connectivity between the hippocampus and the isocortex. Congruently, Plácido and colleagues [54] found that EE elicited differences in the levels of monoamines, cell proliferation and neuronal cytoarchitecture in the hippocampus of the YAC128 HD mouse model. In HD models, such as the R6/1 mouse, EE has been shown to ameliorate deficits in brain-derived neurotrophic factor (BDNF) levels, known to greatly modulate synaptic plasticity [55,56,57]. Upregulating BDNF levels with intracortical injections has been shown to positively influence homeostatic sleep regulation [58] in rats. Therefore, EE may indirectly modulate sleep and circadian rhythms through its propensity to cause structural and functional changes in neuronal circuits.

Mouse models of HD have been pivotal to demonstrate that different forms of enrichment have beneficial influences on cognitive and motor impairments and improve survival. Exposure to stimulating environments, with for instance, regular behavioural testing, wheel running or treadmill, improved motor performance and slowed the decline in cognitive capabilities in several HD animal models, such as the Q140, R6/1 and R6/2 mouse models, as well as a quinolic-induced HD rat model relative to mutants that did not receive any enrichment [55,59,60,61,62]. Even limited EE was enough to slow the decline in motor function and ameliorate the performance of R6/2 and YAC128 mice in the rotarod test [54,63]. Moreover, exposure to EE from a young age significantly assisted in delaying loss of cerebral volume, motor symptoms and the onset of general health decline in both the R6/1 and R6/2 mice [63,64,65]. In addition, exposure to multiple forms of EE prevented the loss of body weight and improved survival in these two models [2,56,61].

### 2.2. Bright & Targeted Wavelengths of Light

Photoentrainment of mammalian circadian rhythms is mediated by a subset of retinal ganglion cells that are intrinsically photosensitive, shortly known as ipRGCs. These cells express the photopigment melanopsin [66], whose functions, including the suppression of melatonin secretion in the pineal gland [67]), play a fundamental role in regulating circadian rhythms [68] as well as pupil dilation, hormonal secretion and affective processes [69,70]. While sensitive to a broad spectrum of light, melanopsin is optimally responsive to blue light with a wavelength of 480 nanometers [71]. As ipRGCs project to the SCN, exposure to bright light results in spikes of neuronal activity in the SCN and increased expression of genes that modulate circadian rhythms, thus aligning the organism’s clock with the timing of the exposure to photoperiods [71,72]. Moreover, changes in the timing of the photoperiod exposures result in shifts in circadian phase [73]. Most importantly and relevant to this review, bright light therapy has been reported to ameliorate circadian deficits in the R6/2 model [48] as well as in clinical research [74]. How bright light improves behavioural rhythms and motor performance is still under investigation.

Wang and colleagues [75] exposed two HD models, the BACHD and Q175 mice, to 6 h of blue-enriched illumination at the start of the light/resting phase for 3 months. While a decline in the power of the locomotor activity rhythms along with increased fragmentation was observed in untreated mutants, treated mice of both strains did not demonstrate such decrements and displayed significantly improved motor performance. In addition to the timing of the exposure, the duration of a light photoperiod can be a major factor in circadian photoentrainment.

The effects and relationships of seasonal variations in day-light length with symptoms presentations have been explored in psychiatric syndromes [76], but scarcely in HD or other neurodegenerative disorders, except for one paper reporting an effect of temperature fluctuations during winter and summer on the age at onset [77]. Humans are not normally considered photoperiodic animals; however, many find themselves living in a long day photoperiod due to the extensive use of artificial lighting, and preclinical evidence suggests that exposure to longer photoperiods might be beneficial. For instance, R6/2 mice held under different photoperiods ultimately showed a significantly delayed disruption in rest-activity rhythms when exposed to a long-day photoperiod (16 h light:8 h dark) relative to mutant mice exposed to a standard light-dark (12 h light:12 h dark, LD) cycle, a short-day cycle (8 h light:16 h dark) or held in constant darkness [78]. Furthermore, R6/2 females exposed to the long-day photoperiod displayed a smaller decrease in the amplitude of rest-activity rhythms over time, along with improved survival and delayed weight loss, suggesting sex divergent effects in the response to light [78]. Differences in the length of the photoperiod (long-days vs. short-days) can result in changes in physiological phenomena that in turn regulate SCN physiology [79,80] and gene expression [81,82,83] with likely implications for regulation of the circadian outputs including the HPA axis [84]. As such, it appears that, as a zeitgeber (“time-giver”), light exposure can influence and align an organism’s circadian rhythms over constant and prolonged periods. The ideal model for this type of work could be a larger mammal in sheep as this species has a robust seasonal physiology and metabolism.

For a variety of practical reasons, mice have become the main driver of preclinical biomedical research; hence, there are numerous transgenic mouse models of HD. Studying neurodegenerative diseases in mouse models comes with some caveats and there are real advances to using large brained, long-lived animals to complement the work done with rodent models [10]. Case in point would be the transgenic sheep model of HD that carries the full-length human cDNA with a CAG repeat of 73 [9]. These animals exhibit disruptions in their activity rhythms [85] as well as in EEG-defined sleep [86], along with alterations in the pattern of melatonin secretion at presymptomatic stages [87].

### 2.3. Scheduled/Time Restricted Feeding

In mammals, a bidirectional relationship exists between circadian rhythms and metabolism, as circadian disruptions predict poor metabolic outcomes [88] and genetically-induced obesity in mice predicts debilitating arrhythmias [89]. The underlying mechanisms of this relationship are not well understood, however, many aspects of metabolism are under circadian control, such as daily variations in glucose levels and tolerance, which appear to be at least in part modulated by the SCN and clock genes [90]. In addition, the SCN appears to indirectly modulate the rhythmicity of peripheral organ clocks involved in metabolic function through efferent targets in the hypothalamus involved in hunger and satiety [91], as well as food-anticipatory behaviour [92]. Additionally, the SCN communicates with organs involved in metabolic function, such as the pancreas [27], regulate metabolic enzymes and production of metabolites through efferent targets and the autonomic nervous system. Roughly 20% of the liver-soluble proteins show strong oscillations that suggest a direct circadian control [26]. Metabolic signals and factors, such as hormones, from peripheral organs and metabolites have functions that affect clocks in the brain, such as those of the pituitary gland, the periventricular nucleus, and the dorsomedial hypothalamus [26,27]. Peripheral clocks can become uncoupled from the SCN, which remains entrained to light rather than food-intake, suggesting the existence of a food-entrainable oscillator (FEO) in the brain, which some argue is within the dorsomedial hypothalamus [93], albeit without consensus. Further research is needed to develop a more comprehensive understanding of the mechanisms that influence the relationship between circadian rhythms and metabolic processes.

While the mechanisms underlying the relationship between circadian rhythms and metabolic function are still under investigation, the benefits of time-restricted feeding (TRF), in which food availability is restricted to a select period of hours without significant alteration of the content or calories of food ingested, are well documented. For instance, Hatori and colleagues [94] found that mice treated with TRF consumed just as many calories from a high-fat diet as those fed *ad libitum* (ALF) yet displayed better motor coordination and were better protected against obesity and other metabolic disorders. Additionally, TRF appears to be most beneficial when the restricted feeding period aligns with the active phase of one’s circadian cycle. In fact, Arble et al. [95] demonstrated that mice fed exclusively during their active/dark phase are better protected from weight gain and obesity than mice whose food availability is restricted to their inactive/light phase. Moreover, obese Histamine H(1)-R knockout mice with arrhythmic feeding patterns displayed improvements in both obesity and related metabolic disorders when solely fed at night [96]. Strikingly, C56BL/6J mice subjected to a TRF regimen that aligned feeding to the active/dark stage had significantly extended lifespans by roughly 35% relative to mice on ALF [97].

For those with HD, the most pertinent benefit of TRF is its potential to establish and entrain circadian rhythms, likely through entraining a FEO that can act independently of the SCN, as dysfunction of the SCN is a characteristic of HD that results in circadian and sleep disruptions that predict a more rapid rate of physical and cognitive decline. By restricting food availability to align with the active phase of R6/2 mice, Maywood et al. [98] demonstrated that activity rhythms are significantly rescued and peak during the time in which the mice anticipate food and that the disintegration of circadian gene expression in hepatic cells are reversed. Additionally, the implication of TRF in male, but not female, BACHD mice significantly corrected inappropriate wake rhythms, improved the temporal patterning of rapid eye-movement (REM) and non-REM (NREM) sleep, and reduced the occurrence of sleep bouts. Moreover, a cosiner analysis found that a higher percentage of both wildtype and BACHD mice treated with TRF had significant diurnal rhythms relative to those with ALF schedules [99]. Furthermore, Northeast et al. [100] showed that, following sleep deprivation between ZT3-6, TRF, but not ALF, augmented slow wave activity, suggesting that this regimen maintains sleep homeostasis. Moreover, the benefits of TRF as a circadian intervention in HD mice models extend to multiple aspects of physical health. For instance, in the R6/2 model, mice treated with TRF displayed improved thermal regulation, locomotor behaviour, and a slower loss of body weight [64]. In both Q175 and BACHD models, the restriction of food availability to solely the peak of the active phase (ZT 15–21) over a period of three or six months resulted in improved locomotor activity, heart rate variability, behavioural sleep patterns and motor improvements that were positively associated with improved circadian outputs [3,4].

The exact mechanisms underlying the benefits of TRF are unknown, however, there are findings that provide insight into possible mechanisms. For instance, deficits in BDNF expression levels, which play a critical role in the modulation of both cell survival and synaptic plasticity [57], are rescued. When circadian patterns of food availability are misaligned, long-term potentiation is inhibited and hippocampal total CREB levels are reduced [101]. Conversely, TRF upregulates BDNF expression [4] and induces structural and functional changes in the hippocampal regions and the septum, which appear to be paralleled by the SCN’s rhythms [102]. Moreover, in HD mouse models, methods that increase the production, release, or signalling of BDNF have resulted in improvements of HD phenotypes [57]. In addition, mice treated with TRF display improved rhythmic cycles in genes associated with autophagy [103] and lower m*HTT* levels associated with the upregulation of autophagy [42], suggesting that autophagy may play a significant role in improvements seen through TRF. There are also indications that TRF downregulates inflammatory signaling and normalizes the daily patterns of expression for genes involved in inflammation [44,103]. Moreover, prolonged periods of fasting induce depletion of hepatic glycogen that results in ketone bodies being the primary modulator of energy metabolism, which some believe benefits those with neurodegenerative disorders [104]. Albeit the wealth of evidence, further research is still needed to pin down some players in the underlying mechanism(s) for more targeted interventions.

### 2.4. Ketogenic Diet

Individuals with HD, like those with other neurodegenerative disorders, suffer from deficits in glucose metabolism in the brain that precede the onset of symptoms and could be directly associated with the severity of the disease [105,106,107,108,109,110,111,112]. Multiple metabolic processes, including glucose uptake and mitochondrial functions, are significantly impaired in all types of neural cells (neurons and glia) in neurodegenerative disorders, along with exacerbated oxidative stress and consequent cell loss in specific brain areas, for instance, the medium spiny neurons in the striatum in HD or the dopaminergic neurons in the substantia nigra in Parkinson’s Disease [104]. As such, the upregulation of alternative energy sources, for instance ketones, could serve as crucial agents in alleviating such “hypometabolism”. This idea is further supported by data in preclinical models of HD in which manipulations aimed at increasing ketone body availability in the circulatory system have been shown to improve survival, slow the loss of body weight, ameliorate deficits in motor symptoms [113,114], and, in a case-study, improve a patient’s quality of life [115]. The use of a high-in-fat and low-in-carbohydrates ketogenic diet (KD), with the goal of artificially upregulating ketone body production and inducing prolonged periods of ketosis, has been proposed and tested in a number of neurodegenerative disorders at both the clinical and preclinical level, including HD [115].

In the R6/2 model, Ruskin et al. [116] reported that the onset of weight loss was significantly delayed in mice fed a KD, while motor symptoms were not further impaired or ameliorated in the mutants. Impaired working memory was reversed in the mutant females but not in the males. Moreover, BACHD mice fed a KD *ad libitum* displayed significantly altered expressions of genes associated with neurodegenerative disorders, inflammation and sleep, along with strongly improved rhythms in activity, increased power and reduced activity onset variability. Conversely, KD had little effects on sleep parameters, albeit the mutants presented with less fragmented sleep, and an advanced sleep onset relative to their counterpart fed with normal chow. Additionally, circadian enhancement was accompanied by improved motor performance on the rotarod and challenging beam tasks [117]. The mechanisms underlying the observed benefits of upregulating ketone body production are unknown, however, there is evidence suggesting that ketone bodies might influence and downregulate the signalling of inflammatory pathways whilst upregulating autophagy [118]. Furthermore, KD has been shown to elicit substantial changes in the composition of the gut microbiota, which has also been suggested to play a role in modulating HD pathogenesis [119]. In accord, we have recently reported that KD evoked drastic changes in the microbiome of BACHD mice, in particular, in the levels of the probiotic *Akkermansia Muciniphila* [117], which has been associated with improved metabolic health and reductions in inflammation [120,121]. A significant increase in the gene expression levels of BDNF was also observed in the BACHD mice, likely mediated by the increase in β-hydroxybutyrate [122] elicited by the KD as compared to mutants on normal chow. Interestingly, the increase in β-hydroxybutyrate was significantly higher in the mutants as compared to wild-types [117]. These findings support the use of KD as neuroprotective and to ameliorate and, perhaps, delay the symptoms and disease progression in HD individuals [115], but further research is still needed.

Quality sleep and properly synchronized sleep/wake cycles are important to maintain whole body homeostatic functions, with devastating consequence in case of malalignment or low sleep quality. Although the sleep and circadian benefits of the KD are not well understood, it has been suggested that it may function via increased activation of orexinergic neurons, involved in feeding, sleep and wakefulness or upregulation of NAD+ and sirtuin, both associated with longevity and implicated in the sleep/wake cycle decay during aging [123,124]. In a recent study using a modified KD containing medium-chain triglycerides to avoid eliciting carbohydrate starvation, it was found that only REM sleep was significantly impacted, with no changes in NREM sleep or wake, as well as in the molecular clock, rhythms in locomotor activity or in core body temperature [125]. Too few studies in humans and animal models presently available to surmise a possible mechanistic view, however, some connections and conjectures can be made since: i. both sleep and KD modulate energy homeostasis and metabolism; ii. improving sleep or using a KD can ameliorate cognitive and metabolic dysfunctions; iii. albeit very few, some studies suggest that slow wave sleep (SWS) is modulated by fasting and associated ketogenic phase, which in turn could mediate some of the KD effects. SWS is important for memory consolidation and cognitive functions and is impacted in individuals with metabolic syndromes and linked to cognitive decline in neurodegenerative disorders and in aging [126,127,128].

## 3. Neuroendocrine Dysfunction in HD

The TRF and KD benefits are likely to be mediated by the hypothalamus and to impact metabolism. These interventions make sense given that HD patients exhibit metabolic changes, including elevated energy expenditure [129], impaired cholesterol- and fatty acid synthesis [130,131], and, importantly, progressive weight loss [132]; along with the fact that many of the non-motor symptoms are consistent with hypothalamic dysfunctions, including sleep and circadian disorders. Pathological features of HD, including metabolic changes, can be reproduced in wild-type mice by the targeted overexpression of m*HTT* in hypothalamic neurons [133,134,135]. Conversely, the selective deletion of m*HTT* from the hypothalamus was sufficient to ameliorate obesity and depressive-like behaviour in BACHD mice [133,136]. Furthermore, significant hypothalamic pathology occurs in the R6/2 model [137,138,139]. The hypothalamus plays a significant role in the regulation of neuroendocrine functions and acts as a major control center to integrate signals from both the nervous and the endocrine system. Consequently, the described HD-driven pathology of the hypothalamus [140] would be expected to generate neuroendocrine dysfunction which might be counteracted and delayed by TRF or KD regimen.

The hormones melatonin and cortisol serve in the communications between the SCN and the peripheral oscillators and are considered indicators of circadian rhythms. Studies have evaluated the effects of TRF on circadian rhythms and the levels of these hormones in both humans and animals, but not in HD models. The pathways involved in the nightly rise in melatonin secretion are relatively well-defined, but human data have been inconsistent and there is still lack of consensus on the impact of HD on its rhythms. Early work reported a phase delay in melatonin secretion, as well as reduced levels in premanifest HD individuals, which worsened with progression to manifest HD [141]; in contrast, Bartlett and colleagues [142] observed no changes in dim light melatonin onset in premanifest HD, even among those with reduced sleep quality. Recent work in the R6/2 mouse, as well as in HD patients, suggests a significant reduction in the expression of Aralkylamine N-acetyltransferase (AANAT), the rate-limiting step enzyme in melatonin biosynthesis [143]. Despite the latter, we have still little or no knowledge on the impact of HD on melatonin rhythmicity in preclinical models since most mouse strains are on the C57BL/6 background that does not produce melatonin, hence, the need of using other organisms to model HD [10,87]. Cortisol, one of the body’s “wake signals”, displays one of the most robust circadian rhythms. Peaking just prior to waking, its release from the adrenal cortex is regulated by the HPA axis, also under circadian control. Evidence of slightly altered or unaltered rhythms in cortisol secretion have been obtained in premanifest and manifest HD individuals [142,144,145], with later stages of HD showing elevated cortisol [146] and a phase advance similar to that observed in aging [145,147]. There has been very little work done on possible disruptions of the circadian rhythm in the HPA axis in HD models. The R6/2 mice exhibit elevated corticosterone levels, and strikingly, modulating corticosterone or glucocorticoids levels can either exacerbate or improve HD symptomatology [148,149,150]. Therefore, interventions aimed at normalizing the disrupted rhythms in melatonin and cortisol are promising avenues for future work [151].

Investigation on the possible involvement of peripheral hormones associated with satiety and appetite (e.g., ghrelin and leptin) [152] can contribute to a more complete understanding of metabolic dysfunctions in HD as well as the beneficial, or deleterious, effects of TRF. The secretion of leptin and ghrelin, respectively from adipose tissue and the stomach, aligns with the feeding schedule and exhibits almost opposite daily rhythms. Preclinical research suggests impairments in the function of these hormones in HD. For instance, the BACHD mouse model shows a development of leptin resistance and selective expression of mHtt in the hypothalamus of FVB/N female mice was sufficient to replicate such dysfunction [133]. Moreover, in the late stage of the disease, R6/2 mice display low levels of ghrelin, resulting in ghrelin-axis deficiencies and impairments of its functions [153]. Despite these preclinical findings, further research is still needed to determine if the basic rhythms of these hormones are impaired in individuals with HD [154,155,156,157,158]. Leptin rhythmic secretion and resistance in wild-type mice have been shown to be driven by the time of feeding [159,160], while TRF elicited the alignment of ghrelin secretion with the mealtime [161]. Moreover, jet-lagged C57BL/6J mice lost diurnal ghrelin rhythms were corrected by TRF [162]. While it is established that ghrelin and leptin rhythms are modulated by the feeding schedule, the effect of TRF on the total levels of these hormones remains unclear and, importantly, unknown in HD models.

## 4. Circadian Desynchrony and Cardiovascular Pathology in HD

Cardiac events, such as heart failure, are highly prevalent in individuals with HD [163]. Expression of a long polyglutamine repeat (>50) in cardiomyocytes caused heart failure in mice [164], whilst selective removal of mHtt from these cells benefited cardiovascular function in BACHD mice and improved their performance in cardiovascular-sensitive motor tasks [165]. A wide range of cardiac defects has been documented in individuals with HD and preclinical HD models, in particular those strongly associated with dysautonomia, such as abnormal heart rate variability [165,166,167,168,169]. While the cause of these cardiac events is unknown, there are indications that SCN-related disruptions in circadian rhythms lead to dysautonomia in HD. The SCN projects to the paraventricular nucleus (PVN) a regulator of the autonomic output that modulates cardiovascular rhythmicity. Additionally, the master clock is involved in the modulation of the HPA axis, also implicated in the regulation of cardiovascular functions [16]. Therefore, it is likely that the disruption in the molecular clock and SCN functions, along with the described degeneration of brainstem nuclei observed in HD, strongly weaken autonomic nervous system functions and rhythmicity, promoting cardiac dysrhythmia and other malevents [168,170,171,172]. The impaired brain-heart communications and heart malfunction will perhaps exacerbate disease progression, and dramatically impact the quality of life of HD individuals, hence resynchronization of the central and peripheral molecular clocks with chrono-based therapies has the potential to improve heart health.

## 5. Translation into Clinical Practice: Efficacy and Limitations

The goal of research in preclinical models is to provide information about mechanisms and interventions that can be translated into clinical practice. As such, when looking to apply interventions that show promise in preclinical models, one must consider the efficacy, feasibility, and risks of the intervention, with particular consideration to the prospect of physical or mental harm that patients may experience. Several protocols of physical exercise have been tried in early-to-moderate stage HD patients, with such interventions being perceived as feasible and well-tolerated by patients. Additionally, both physical and cognitive training regimens appear to benefit motor function, cognition, mood, sleep architecture, and quality of life in HD patients [173,174,175,176,177]. There are also indications that a physically and mentally active lifestyle prior to symptom onset may greatly delay the onset and decrease the severity of HD [46]. Moreover, in a case study, a 41-year-old patient who stayed on a time-restricted ketogenic diet for 48 weeks presented with improvements in multiple factors of physical and mental well-being [115]. Clinical studies have also found that bright light therapies may benefit sleep quality in dementia cohorts [178]. Still, it should be noted that the exact amplitude of the benefits of circadian interventions on HD pathophysiology are still uncertain, although counteracting effects on the associated neurodegenerative events could be predicted, followed by delayed loss of cognitive and motor functions.

Despite promising evidence, we must also consider that, in clinical practice, ketogenic diets are often difficult to adhere to and are associated with adverse physical health effects and the emotional dissatisfaction of the patients [104,128]. Moreover, Skillings et al. [64] found that, whilst EE and TRF were individually beneficial for the R6/2 mice, the simultaneous use of the two interventions resulted in earlier mortality for the mice. In addition, degeneration of ipRGCs and decreased expression of melanopsin in HD appear to occur prior to the onset of motor symptoms [179], suggesting that bright light therapies may only be effective in the very early stages of disease progression. It must also be stated that, likely due to the widespread access to artificial bright lights in homes, adherence to bright light therapies is often poor [178]. As such, efforts are being made to implement tunable home-based white lighting systems which serve to increase the regularity of exposure to bright light and improve entrainment for those with neurodegenerative disorders [180], however, this intervention is at a rudimentary stage of development, with further research needed to determine its efficacy and feasibility.

## 6. Concluding Remarks

In this current age of research on HD, there is no cure or pharmacological treatment that significantly delays the progression of the disease. As such, interventions that can improve the quality of life of patients may present a path forward for the near-future of clinical practice. In preclinical models of HD, the application of circadian interventions aimed at improving synchronization and limiting disturbances to the sleep/wake cycle has been shown to be effective in delaying or improving motor and cognitive deficits, and certain clinical studies have supported the efficacy of their translation. Whilst these are promising provisions, it may yet be too soon to apply such chrono-interventions to organized clinical practice. Interventions are rarely lacking in risks that may negatively affect the quality of life of patients with respect to their physical or mental well-being. Thus, it must be considered that little is known of the mechanisms of the observed benefits of circadian interventions, and even less is known about the risks of applying one or multiple interventions simultaneously in patient populations. Furthermore, a greater understanding of how to apply these interventions to guarantee that both adherence and patient satisfaction are feasible is a necessary component for such translation. Therefore, more research is required to understand the mechanisms and risks of these interventions as well as develop a standard of care for their use in clinical settings.

## Figures and Tables

**Figure 1 biomedicines-12-01777-f001:**
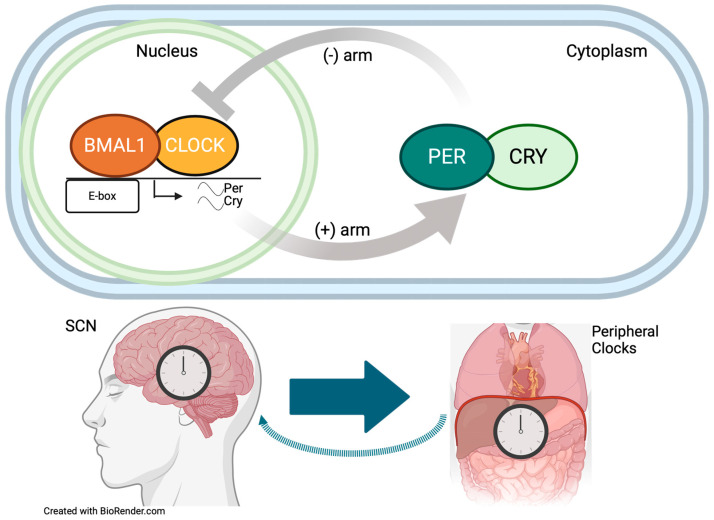
Circadian rhythms are driven by a molecular feedback loop in which there are near-24-h oscillations in the expression of core clock proteins. The oscillations are initiated by the “positive arm”, when dimers of the transcription factors BMAL1 and CLOCK bind to the E-box consensus sequence to upregulate the expression of the proteins PER and CRY. These translocate and accumulate in the cytoplasm, driving the “negative arm” to interfere with the ability of BMAL1 and CLOCK to bind to DNA. Subsequently, the expression levels of PER and CRY gradually decline, allowing BMAL1 and CLOCK to re-initiate this cycle. The suprachiasmatic nucleus (SCN), found in the anterior hypothalamus, is recognized as the “master clock”, which orchestrates the rhythms of the molecular clocks in all neural and peripheral tissues. Major peripheral physiological processes, for instance in the liver, are subject to circadian regulation and display circadian fluctuations. Desynchrony of the circadian timing system can result in deleterious effects on health and exacerbate existing malconditions.

**Figure 2 biomedicines-12-01777-f002:**
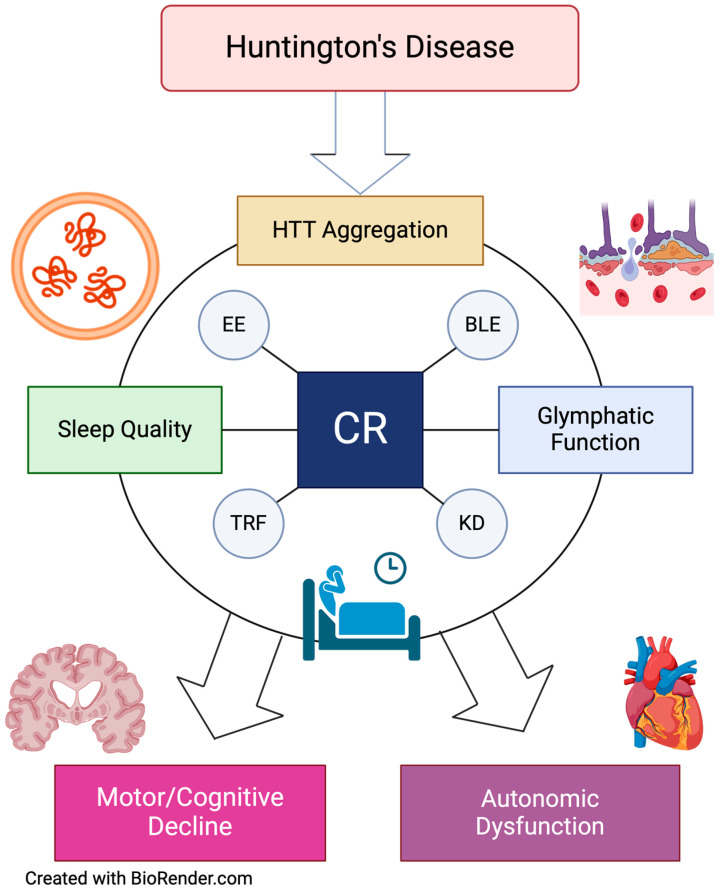
Huntington’s disease (HD) is a neurodegenerative disorder caused by an extended CAG polyglutamate repeat in the Huntingtin (HTT) gene that results in protein aggregation and the disruption of neural function. Sleep disorders are common for individuals with HD and sleep quality holds a bi-directional relationship with the prognosis of HD, in which a malfunctioning glymphatic system could perhaps play a significant role. Diminished sleep quality could be directly associated with greater amounts of HTT aggregation, reduced glymphatic clearance, autonomic dysfunction, and a higher rate of motor and cognitive decline. Through behavioural, circadian-based interventions, the sleep quality of HD patients can be improved, which may play a vital role in slowing the progression of the disease. CR: circadian rhythms; EE: enriched environment; BLE: blue light exposure; TRF: time restricted feeding; KD: ketogenic diet.

## Data Availability

Not applicable.

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
