# Peer review of "Circadian Interventions in Preclinical Models of Huntington’s Disease: A Narrative Review"

_biomedicines, 2024, doi:10.3390/biomedicines12081777_

Round 1

Reviewer 1 Report

Comments and Suggestions for Authors

In the paper “Circadian Interventions in Preclinical Models of Huntington’s Disease: A Narrative Review”, by Dell’Angelica et al. the authors attempt to emphasize the novel approach in the treatment of Huntington’s disease that is currently incurable. They carefully explain the treatments that affect the circadian rhythm ameliorating the pathology in the animal models of Huntington’s disease.

The paper is interesting and innovative, suggesting future paths of research that can eventually benefit the patients. Thus, it merits publication.

I have only one comment –

Line 213: The sentence “While the mechanisms underlying the relationship between circadian rhythms and metabolic function, the benefits of time-restricted feeding (TRF), in which food availability is restricted to a select period of hours without significant alteration of the content or calories of food ingested, are well documented.” is not complete.

Author Response

Reviewer 1

In the paper “Circadian Interventions in Preclinical Models of Huntington’s Disease: A Narrative Review”, by Dell’Angelica et al. the authors attempt to emphasize the novel approach in the treatment of Huntington’s disease that is currently incurable. They carefully explain the treatments that affect the circadian rhythm ameliorating the pathology in the animal models of Huntington’s disease. 

The paper is interesting and innovative, suggesting future paths of research that can eventually benefit the patients. Thus, it merits publication.

I have only one comment – 

Line 213: The sentence “While the mechanisms underlying the relationship between circadian rhythms and metabolic function, the benefits of time-restricted feeding (TRF), in which food availability is restricted to a select period of hours without significant alteration of the content or calories of food ingested, are well documented.” is not complete. 

We thank the reviewer for his time and for picking up this mistake. We have corrected the sentence. Line 246-249:

“While the mechanisms underlying the relationship between circadian rhythms and metabolic function are still under investigation, the benefits of time-restricted feeding (TRF), in which food availability is restricted to a select period of hours without significant alteration of the content or calories of food ingested, are well documented.”

Reviewer 2 Report

Comments and Suggestions for Authors

The manuscript titled "Circadian Interventions in Preclinical Models of Huntington's Disease: A Narrative Review" presents a comprehensive analysis of preclinical studies investigating circadian-based interventions in Huntington's Disease (HD). While the paper provides detailed case studies and acknowledges the limitations and challenges in translating circadian interventions to clinical applications, some areas require improvement for publish:

1. Model-Matching: It is important to ensure that intervention results are linked to specific HD models, such as the YAC128 HD mouse model. Care should be taken to avoid instances where this information is omitted or unclear.

2. Standardize Paragraph Formatting: Consistency in paragraph formatting, including the use of indentation for the first line, should be implemented throughout the manuscript to enhance readability.

3. Clarify Abbreviations: Upon their first occurrence, abbreviations should be explained in full. For example, the suprachiasmatic nucleus (SCN) should be defined upon its initial mention to aid readers' understanding.

4. Strengthen "Preclinical Evidence" Summary: The "Preclinical Evidence" section should provide a more robust and concise summary of the findings from the numerous studies discussed, highlighting the key insights and their significance.

5. Unify Reference List Format: The reference list should be standardized to a single format, as it currently includes more than two different formats.

Author Response

Reviewer 2

The manuscript titled "Circadian Interventions in Preclinical Models of Huntington's Disease: A Narrative Review" presents a comprehensive analysis of preclinical studies investigating circadian-based interventions in Huntington's Disease (HD). While the paper provides detailed case studies and acknowledges the limitations and challenges in translating circadian interventions to clinical applications, some areas require improvement for publish:

  1. Model-Matching: It is important to ensure that intervention results are linked to specific HD models, such as the YAC128 HD mouse model. Care should be taken to avoid instances where this information is omitted or unclear.

We thank the reviewer and now clearly state the model/models used with each intervention. 

  1. Standardize Paragraph Formatting: Consistency in paragraph formatting, including the use of indentation for the first line, should be implemented throughout the manuscript to enhance readability.

We apologies for this issue and have corrected the formatting errors. 

  1. Clarify Abbreviations: Upon their first occurrence, abbreviations should be explained in full. For example, the suprachiasmatic nucleus (SCN) should be defined upon its initial mention to aid readers' understanding.

Thank you for pointing this out, we have checked and make sure that all the abbreviations are defined the first time they are used.

  1. Strengthen "Preclinical Evidence" Summary: The "Preclinical Evidence" section should provide a more robust and concise summary of the findings from the numerous studies discussed, highlighting the key insights and their significance.

We have tried to follow the reviewer recommendation, and hope they will find this sections improved.

  1. Reference List Format: The reference list should be standardized to a single format, as it currently includes more than two different formats.

We have standardized the reference list and now adheres to the journal guidelines. 

Reviewer 3 Report

Comments and Suggestions for Authors

The paper submitted for review addresses the bidirectional interactions between the biological clock and Huntington's disease (HD) and the prognosis of the clinical application of the therapies presented. As reported by the authors, this is a relatively poorly explored topic and it is therefore useful to compile the scientific data to date in the form of a review paper.

The paper is compelling and has been prepared in a thorough and careful manner, has a clear layout, consists of 4 main chapters and conclusion remarks. The authors have appropriately selected references to the subject matter of the work. The publication is enriched by two figures that have been properly described.

Despite my good evaluation of the paper, I have a few comments below and ask the authors to address them in the publication.

1/. the publication lacks a chapter on disrupted hormone secretion rhythms in HD patients/animal models. Apart from the mention of melatonin in chapter 2.2, there are no examples of other hormones e.g. cortisol, pituitary hormones or leptin and ghrelin in the case of food intake rhythms and, of course, orexins, which are crucial for the sleep-wake cycle.

2/. Is there experimental data on the effect of seasons on the course of HD in patients or in seasonal animal models? Please comment on this in subchapter 2.2.

3/. There is no specification in the paper of the repeatedly used term ''progression of HD'' or ''slowing progression of HD''. As a reader, I would like to know how or what the prognosis is that the proposed therapy will affect the severity of the disease, e.g. whether by improving the mental state and higher cerebral functions, the efficiency of the musculoskeletal system, the tension and strength of the muscles, the coherence of movements, the functions of the autonomic nervous system, etc.

4/. When referring to animal models in neurodegenerative diseases, it is good to refer to large animal models in addition to rodents, as these studies more accurately reflect the progression of human neurodegeneration compared to rodent models in which accelerated forms of the neurodegenerative disorders are induced to match their only 2-year lifespan. Whereas large animals such as sheep live an average of 12 years.

As far as editorial comments are concerned, please explain the HPA axis abbreviation (line 114) while in line 356 remove the explanation.

Author Response

Reviewer 3

The paper submitted for review addresses the bidirectional interactions between the biological clock and Huntington's disease (HD) and the prognosis of the clinical application of the therapies presented. As reported by the authors, this is a relatively poorly explored topic and it is therefore useful to compile the scientific data to date in the form of a review paper.

The paper is compelling and has been prepared in a thorough and careful manner, has a clear layout, consists of 4 main chapters and conclusion remarks. The authors have appropriately selected references to the subject matter of the work. The publication is enriched by two figures that have been properly described.

Despite my good evaluation of the paper, I have a few comments below and ask the authors to address them in the publication.

  1. the publication lacks a chapter on disrupted hormone secretion rhythms in HD patients/animal models. Apart from the mention of melatonin in chapter 2.2, there are no examples of other hormones e.g. cortisol, pituitary hormones or leptin and ghrelin in the case of food intake rhythms and, of course, orexins, which are crucial for the sleep-wake cycle.

We have added a paragraph entitled “3. Neuroendocrine dysfunction in HD” and reported the work that has been completed in HD preclinical and clinical studies. As highlighted in the paragraph, there is a surprising lack of information about the impact of HD on hormonal secretion in animal models, and there is a lack of consensus on the findings in premanifest and manifest HD individuals. 

2/. Is there experimental data on the effect of seasons on the course of HD in patients or in seasonal animal models? Please comment on this in subchapter 2.2.

Interesting question. Photoperiod drives major changes in behaviour and neuroendocrine systems in animal models and these changes are reflected in alterations of the SCN physiology. The closest and only data available are on the effects of changes in temperature through winter and summer on the age at onset. Brackenridge C. J. (1974) observed a decrease in age at onset as the temperature increased through winter and summer, with the winter temperature exerting the stronger effect compared to summer

In winter: age at onset was 34 at 10-19degree F and declined to 26 when the temp reached 50-59F.

In Summer: age at onset was about 34-35 at 50-59F and 31-32 at 80-89F.

We have added this info and comments, see lines 170-190.

“The effects and relationships of seasonal variations in day-light length with symptoms presentations have been explored in psychiatric syndromes (Zhang & Volkow, 2023), but scarsely in HD or other neurodegenerative disorders, except for one paper reporting an effect of temperature fluctuations during winter and summer on the age at onset (Brackenridge, 1974). Humans are not normally considered photoperiodic animals; however, many find themselves living in a long day photoperiod due to the extensive use of artificial lighting, and preclinical evidence suggests that exposure to longer photoperiods might be beneficial……………”

Brackenridge C. J. (1974). Effect of climatic temperature on the age of onset of Huntington's chorea. Journal of neurology, neurosurgery, and psychiatry, 37(3), 297–301. https://doi.org/10.1136/jnnp.37.3.297

3/. There is no specification in the paper of the repeatedly used term ''progression of HD'' or ''slowing progression of HD''. As a reader, I would like to know how or what the prognosis is that the proposed therapy will affect the severity of the disease, e.g. whether by improving the mental state and higher cerebral functions, the efficiency of the musculoskeletal system, the tension and strength of the muscles, the coherence of movements, the functions of the autonomic nervous system, etc.

The reviewer points out one of the key questions in this literature. Several interventions that improve behaviour have been tested, but it is not yet know the impact on the core pathology of the disease, although the results reported by Phillips et al, 2022 seem promising. We have highlighted a series of sentences and paragraphs in which the expectations/speculations on the possible outcomes are reported as well as the limitations. See in particular lines 85-88, 466-469.

4/. When referring to animal models in neurodegenerative diseases, it is good to refer to large animal models in addition to rodents, as these studies more accurately reflect the progression of human neurodegeneration compared to rodent models in which accelerated forms of the neurodegenerative disorders are induced to match their only 2-year lifespan. Whereas large animals such as sheep live an average of 12 years. 

We thank the reviewer for this suggestion and agree that the inclusion of the findings obtained with the HD sheep model are an important aspect of preclinical research to review. Please see lines 48-50 and 191-198,  

As far as editorial comments are concerned, please explain the HPA axis abbreviation (line 114) while in line 356 remove the explanation.

Thank you for pointing this out. We have made these corrections.